# Confidence Sets for Statistical Classification (II): Exact Confidence Sets

**Wei Liu [1,\*]**, **Frank Bretz [2]** and **Anthony J. Hayter [3]**

1. S3RI, School of Mathematics University of Southampton, Southampton SO17 1BJ, UK
2. Novartis Pharma AG, 4002 Basel, Switzerland; frank.bretz@novartis.com
3. Department of Statistics and Operations Technology, University of Denver, Denver, CO 80208, USA; Anthony.hayter@du.edu
* Correspondence: w.liu@maths.soton.ac.uk

**Abstract:** Classification has applications in a wide range of fields including medicine, engineering, computer science and social sciences among others. Liu et al. (2019) proposed a confidence-set-based classifier that classifies a future object into a single class only when there is enough evidence to warrant this, and into several classes otherwise. By allowing classification of an object into possibly more than one class, this classifier guarantees a pre-specified proportion of correct classification among all future objects. However, the classifier uses a conservative critical constant. In this paper, we show how to determine the exact critical constant in applications where prior knowledge about the proportions of the future objects from each class is available. As the exact critical constant is smaller than the conservative critical constant given by Liu et al. (2019), the classifier using the exact critical constant is better than the classifier by Liu et al. (2019) as expected. An example is provided to illustrate the method.

**Keywords:** classification; confidence level; confidence set; coverage frequency; statistical inference

## 1. Introduction

Classification has applications in a wide range of fields including medicine, engineering, computer science and social sciences among others. For overviews, the reader is referred to the books by [1–5]. In the recent paper, Liu et al. (2019) [6] proposed a new classifier based on confidence sets. It constructs a confidence set for the the unknown parameter $c$, the true class of each future object, and classifies the object as belonging to the set of classes given by the confidence set. Hence, this approach classifies a future object into a single class only when there is enough evidence to warrant this, and into several classes otherwise. By allowing classification of an object into potentially more than one class, this classifier guarantees a pre-specified proportion of correct classification among all future objects with a pre-specified confidence $\gamma$ about the randomness in the training data based on which the classifier is constructed.

However, the classifier of Liu et al. (2019) uses a conservative critical constant $\lambda$ and so the resultant confidence sets may be larger than necessary. The purpose of this paper is to determine the exact critical constant $\lambda$ and therefore to improve the classifier of Liu et al. (2019) in situations where one has prior knowledge about the proportions of the (infinite) future objects belonging to the $k$ possible classes.

The layout of the paper is as follows. Section 2 gives a very brief review of the classifier of Liu et al. (2019), and then considers the determination of the exact critical constant $\lambda$ under the additional knowledge/assumption given above. An illustrative example is given in Section 3 to demonstrate the advantage of the improved classifier proposed in this paper when the additional

assumption holds. Section 4 contains conclusions and discussions. Finally, some mathematical details are provided in the Appendix A. As the same setting and notation as in the work by Liu et al. (2019) are used, it is recommended to read this paper in conjunction with the one by Liu et al. (2019).

## 2. Methodology

### 2.1. Methodology

Let the $p$-dimensional data vector $\mathbf{x}_l = (x_{l1}, \ldots, x_{lp})^T$ denote the feature measurement on an object from the $l$th class, which has multivariate normal distribution $N(\boldsymbol{\mu}_l, \Sigma_l)$, $l = 1, \ldots, k$; here, $k$ denotes the total number of classes, which is a known number. The available training dataset is given by $\mathcal{T} = \{\mathbf{x}_{l1}, \ldots, \mathbf{x}_{ln_l}; l = 1, \ldots, k\}$, where $\mathbf{x}_{l1}, \ldots, \mathbf{x}_{ln_l}$ are i.i.d. observations from the $l$th class with distribution $N(\boldsymbol{\mu}_l, \Sigma_l)$, $l = 1, \ldots, k$. The classification problem is to make inference about $c$, the true class of a future object, based on the feature measurement $\mathbf{y} = (y_1, \ldots, y_p)^T$ observed on the object, which is only known to belong to one of the $k$ classes and so follows one of the $k$ multivariate normal distributions. In statistical terminology, $c$ is the unknown parameter of interest that takes a possible value in the simple parameter space $C = \{1, \ldots, k\}$. We emphasize that $c$ is treated as non-random in both the work of Liu et al. (2019) and here.

A classifier that classifies an object with measurement $\mathbf{y}$ into one single class in $C = \{1, \ldots, k\}$ can be regarded as a point estimator of $c$. The classifier of Liu et al. (2019) provides a set $\mathcal{C}_{\mathcal{T}}(\mathbf{y}) \subseteq C$ as plausible values of $c$. Depending on $\mathbf{y}$ and the training dataset $\mathcal{T}$, $\mathcal{C}_{\mathcal{T}}(\mathbf{y})$ may contain only a single value, in which case $\mathbf{y}$ is classified into one single class given by $\mathcal{C}_{\mathcal{T}}(\mathbf{y})$. When $\mathcal{C}_{\mathcal{T}}(\mathbf{y})$ contains more than one value in $C$, $\mathbf{y}$ is classified as possibly belonging to the several classes given by $\mathcal{C}_{\mathcal{T}}(\mathbf{y})$. Hence, in statistical terms, the classifier uses the confidence set approach. The inherent advantage of the confidence set approach over the point estimation approach is the guaranteed $1 - \alpha$ proportion of confidence sets that contain the true classes.

Specifically, the set $\mathcal{C}_{\mathcal{T}}(\mathbf{y}) \subseteq C$ was constructed by Liu et al. (2019) as

$$\mathcal{C}_{\mathcal{T}}(\mathbf{y}) = \left\{ l \in C : (\mathbf{y} - \hat{\boldsymbol{\mu}}_l)^T \hat{\Sigma}_l^{-1} (\mathbf{y} - \hat{\boldsymbol{\mu}}_l) \leq \lambda \right\}, \tag{1}$$

where $\hat{\boldsymbol{\mu}}_l = \frac{1}{n_l} \sum_{m=1}^{n_l} \mathbf{x}_{lm}$ and $\hat{\Sigma}_l = \frac{1}{n_l - 1} \sum_{m=1}^{n_l} (\mathbf{x}_{lm} - \hat{\boldsymbol{\mu}}_l)(\mathbf{x}_{lm} - \hat{\boldsymbol{\mu}}_l)^T$, $l = 1, \ldots, k$, are, respectively, the usual estimators of the unknown $\boldsymbol{\mu}_l$ and $\Sigma_l$ based on the training dataset $\mathcal{T} = \{\mathbf{x}_{l1}, \ldots, \mathbf{x}_{ln_l}; l = 1, \ldots, k\}$, and $\lambda$ is a suitably chosen critical constant whose determination is considered next. The intuition behind the definition of $\mathcal{C}_{\mathcal{T}}(\mathbf{y})$ in Equation (1) is that a future object $\mathbf{y}$ is likely to be from class $l$ if and only if $(\mathbf{y} - \hat{\boldsymbol{\mu}}_l)^T \hat{\Sigma}_l^{-1} (\mathbf{y} - \hat{\boldsymbol{\mu}}_l) \leq \lambda$.

Note that the proportion of the future confidence sets $\mathcal{C}_{\mathcal{T}}(\mathbf{y}_j)$ $(j = 1, 2, \ldots)$ that include the true classes $c_j$ of $\mathbf{y}_j$ $(j = 1, 2, \ldots)$ is given by $\liminf_{N \to \infty} \frac{1}{N} \sum_{j=1}^{N} I_{\{c_j \in \mathcal{C}_{\mathcal{T}}(\mathbf{y}_j)\}}$. Thus, it is desirable that

$$\liminf_{N \to \infty} \frac{1}{N} \sum_{j=1}^{N} I_{\{c_j \in \mathcal{C}_{\mathcal{T}}(\mathbf{y}_j)\}} \geq 1 - \alpha \tag{2}$$

where $1 - \alpha$ is a pre-specified large (close to 1) proportion, e.g., 0.95. While the constraint in Equation (2) is difficult to deal with, Liu et al. (2019) showed that a sufficient condition for guaranteeing Equation (2) is

$$\inf_{c_j \in C} E_{\mathbf{y}_j | \mathcal{T}} I_{\{c_j \in \mathcal{C}_{\mathcal{T}}(\mathbf{y}_j)\}} \geq 1 - \alpha \tag{3}$$

where $E_{\mathbf{y}_j | \mathcal{T}}$ denotes the conditional expectation with respect to the random variable $\mathbf{y}_j$ conditioning on the training dataset $\mathcal{T}$ (or, equivalently, $\{(\hat{\boldsymbol{\mu}}_1, \hat{\Sigma}_1), \ldots, (\hat{\boldsymbol{\mu}}_k, \hat{\Sigma}_k)\}$).

Since the value of the expression on the left hand side of the inequality in Equation (3) (and in Equation (2) as well) depends on $\mathcal{T}$ and $\mathcal{T}$ is random, the inequality in Equation (3) cannot

be guaranteed for each observed $\mathcal{T}$. We therefore guarantee Equation (3) with a large (close to 1) probability $\gamma$ with respect to the randomness in $\mathcal{T}$:

$$P_{\mathcal{T}}\left\{\inf_{c_j \in C} E_{\mathbf{y}_j|\mathcal{T}} I_{\left\{c_j \in \mathcal{C}_{\mathcal{T}}(\mathbf{y}_j)\right\}} \geq 1 - \alpha\right\} = \gamma, \tag{4}$$

which in turn guarantees that

$$P_{\mathcal{T}}\left\{\liminf_{N \to \infty} \frac{1}{N}\sum_{j=1}^{N} I_{\left\{c_j \in \mathcal{C}_{\mathcal{T}}(\mathbf{y}_j)\right\}} \geq 1 - \alpha\right\} \geq \gamma. \tag{5}$$

Computer code in $R$ was provided by Liu et al. (2019) to compute the $\lambda$ that solves Equation (4), which allows the confidence sets $\mathcal{C}_{\mathcal{T}}(\mathbf{y}_j)$ in Equation (1) to be constructed for each future object.

The interpretation of Equations (5) and (6) below is that, based on one observed training dataset $\mathcal{T}$, one constructs confidence sets $\mathcal{C}_{\mathcal{T}}(\mathbf{y}_j)$ for the $c_j$s of all future $\mathbf{y}_j$ ($j = 1, 2, \cdots$) and claims that at least $1 - \alpha$ proportion of these confidence sets do contain the true $c_j$s. Then, we are $\gamma$ confident with respect to the randomness in the training dataset $\mathcal{T}$ that the claim is correct.

A natural question is how to find the exact critical constant $\lambda$ that solves the equation

$$P_{\mathcal{T}}\left\{\liminf_{N \to \infty} \frac{1}{N}\sum_{j=1}^{N} I_{\left\{c_j \in \mathcal{C}_{\mathcal{T}}(\mathbf{y}_j)\right\}} \geq 1 - \alpha\right\} = \gamma \tag{6}$$

which is an improvement to the conservative $\lambda$ that solves Equation (4) as given by Liu et al. (2019). Next, we show how to find the exact critical constant $\lambda$ under an additional assumption which is satisfied in some applications.

Assume that, among the $N$ future objects that need to be classified, $N_l$ objects are actually from the $l$th class with the distribution $N(\boldsymbol{\mu}_l, \Sigma_l)$, $l = 1, \ldots, k$. The additional assumption we make is that

$$\lim_{N \to \infty} \frac{N_l}{N} = r_l, \;\; l = 1, \ldots, k \tag{7}$$

where the $r_l$s are assumed to be known constants in the interval $[0, 1]$. Intuitively, this assumption means that we know the proportions of the future objects that belong to each of the $k$ classes, even though we do not know the true class of each individual future object.

The assumption in Equation (7) is reasonable in some applications. For example, when screening for a particular disease among a specific population for preventive purpose, there are $k = 2$ classes: having the disease ($l = 1$) or not having the disease ($l = 2$). If we know the prevalence of the disease, $d$, in the overall population, then $r_1 = d$ and $r_2 = 1 - d$, even though we do not know whether an individual subject has the disease or not.

It is shown in the Appendix A that, under the assumption in Equation (7), Equation (6) is equivalent to

$$P_{\mathbf{u}_l, \{\mathbf{v}_{lm}\}}\left\{\sum_{l=1}^{k} r_l P_{\mathbf{w}_l \mid \mathbf{u}_l, \{\mathbf{v}_{lm}\}}\left\{(\mathbf{w}_l - \mathbf{u}_l)^T \left(\frac{1}{n_l - 1}\sum_{m=1}^{n_l - 1} \mathbf{v}_{lm}\mathbf{v}_{lm}^T\right)^{-1}(\mathbf{w}_l - \mathbf{u}_l) \leq \lambda\right\} \geq 1 - \alpha\right\} = \gamma \tag{8}$$

where

$$\mathbf{w}_l \sim N(\mathbf{0}, I_p), \;\; \mathbf{u}_l \sim N(\mathbf{0}, I_p/n_l), \;\; \mathbf{v}_{lm} \sim N(\mathbf{0}, I_p), \, m = 1, \cdots, n_l - 1 \tag{9}$$

and all the $\mathbf{w}_l$s, $\mathbf{u}_l$s and $\mathbf{v}_{lm}$s are independent, $P_{\mathbf{w}_l \mid \mathbf{u}_l, \{\mathbf{v}_{lm}\}}\{\cdot\}$ denotes the conditional probability about $\mathbf{w}_l$ conditioning on $(\mathbf{u}_l, \{\mathbf{v}_{lm}\})$, and $P_{\mathbf{u}_l, \{\mathbf{v}_{lm}\}}\{\cdot\}$ denotes the probability about $(\mathbf{u}_l, \{\mathbf{v}_{lm}\})$.

## 2.2. Algorithm for Computing the Exact $\lambda$

We now consider how to compute the critical constant $\lambda$ that solves Equation (8). Similar to Liu et al. (2019), this is accomplished by simulation in the following way. From the distributions given in Equation (9), in the $s$th repeat of simulation, $s = 1, \ldots, S$, generate independent

$$\mathbf{u}_l^s \sim N(\mathbf{0}, I_p/n_l), \quad \mathbf{v}_{l1}^s, \ldots, \mathbf{v}_{l(n_l-1)}^s \sim N(\mathbf{0}, I_p); \quad l = 1, \ldots, k.$$

and find the $\lambda = \lambda_s$ so that

$$\sum_{l=1}^{k} r_l P_{\mathbf{w}_l \mid \mathbf{u}_l^s, \{\mathbf{v}_{lm}^s\}} \left\{ (\mathbf{w}_l - \mathbf{u}_l^s)^T \left( \frac{1}{n_l - 1} \sum_{m=1}^{n_l-1} \mathbf{v}_{lm}^s \mathbf{v}_{lm}^{s\,T} \right)^{-1} (\mathbf{w}_l - \mathbf{u}_l^s) \leq \lambda_s \right\} = 1 - \alpha. \tag{10}$$

Repeat this $S$ times to get $\lambda_1, \ldots, \lambda_S$ and order these as $\lambda_{[1]} \leq \ldots \leq \lambda_{[S]}$. It is well known (cf. [7]) that $\lambda_{[\gamma S]}$ converges to the required critical constant $\lambda$ with probability one as $S \to \infty$. Hence, $\lambda_{[\gamma S]}$ is used as the required critical constant $\lambda$ for a large $S$ value, e.g., 10,000.

To find the $\lambda_s$ in Equation (10) for each $s$, we use simulation in the following way. Generate independent random vectors $\{\mathbf{w}_{lq} : q = 1, \ldots, Q; l = 1, \ldots, k\}$ from $N(\mathbf{0}, I_p)$, where $Q$ is the number of simulations for finding $\lambda_s$. For each given value of $\lambda_s > 0$, the expression on the left-side of Equation (10) can be computed by approximating each of the $k$ probabilities involved using the corresponding proportions out of the $Q$ simulations. It is also clear that this expression is monotone increasing in $\lambda_s$. Hence, the $\lambda_s$ that solves Equation (10) can be found by using a searching algorithm; for example, the bi-section method is used in our R code. To approximate reasonably accurately the probabilities with the proportions, a large $Q$ value, e.g., 10,000, should be used.

It is noteworthy from Equations (8) and (9) that $\lambda$ depends only on $\gamma, \alpha, p, k, n_1, \ldots, n_k, r_1, \ldots, r_k$ (and the numbers of simulations $S$ and $Q$, which determine the numerical accuracy of $\lambda$ due to simulation randomness). It is also worth emphasizing that only one $\lambda$ needs to be computed based on the observed training dataset $\mathcal{T}$, which is then used for constructing the confidence sets $\mathcal{C}_{\mathcal{T}}(\mathbf{y}_j)$ and classifying accordingly all future objects.

It is expected that larger values of $S$ and $Q$ will produce more accurate $\lambda$ value, one can use the method discussed by Liu et al. (2019) to assess how the accuracy of $\lambda$ depends on the values of $S$ and $Q$. Similar to the work by Liu et al. (2019), it is recommended to set $S = 10,000$ and $Q = 10,000$ for reasonable computation time and accuracy of $\lambda$ due to simulation randomness.

## 3. An Illustrative Example

As in the work of Liu et al. (2019), the famous `iris` dataset introduced by Fisher (1936) [8] is used in this section to illustrate the method proposed in this paper. The dataset contains $k = 3$ classes representing the three species/classes of Iris flowers (1 = setosa; 2 = versicolor; and 3 = virginica), and has $n_i = 50$ observations from each class in $\mathcal{T}$. Each observation gives the measurements (in centimeters) of the four variables: sepal length and width, and petal length and width.

We focus on the case that only the first two measurements, sepal length and width, are used for classification in order to easily illustrate the method since the acceptance sets $\mathcal{A}_l = \left\{ \mathbf{y} \in R^p : (\mathbf{y} - \hat{\boldsymbol{\mu}}_l)^T \hat{\Sigma}_l^{-1} (\mathbf{y} - \hat{\boldsymbol{\mu}}_l) \leq \lambda \right\}$, $l = 1, 2, 3$ are two-dimensional and thus can be easily plotted in this case. Based on the fifty observations on $p = 2$ measurements from each of the three classes, the $\hat{\boldsymbol{\mu}}_l$ and $\hat{\Sigma}_l$ were given by Liu et al. (2019).

For $\alpha = 5\%$ and $\gamma = 95\%$, the critical constant $\lambda$ that solves Equation (4) was computed by Liu et al. (2019) to be $\lambda_{con} = 9.175$ using $S = 10,000$ and $Q = 10,000$. The corresponding acceptance sets, based on which the confidence set $\mathcal{C}_{\mathcal{T}}(\mathbf{y})$ in Equation (1) can be constructed directly (cf. [6]), are given by

$$\mathcal{A}_l^{con} = \left\{ \mathbf{y} \in R^p : (\mathbf{y} - \hat{\boldsymbol{\mu}}_l)^T \hat{\Sigma}_l^{-1} (\mathbf{y} - \hat{\boldsymbol{\mu}}_l) \leq \lambda_{con} \right\}, \ l = 1, 2, 3$$

and plotted in Figure 1 by the dotted ellipsoidal region centered at $\hat{\mu}_l$, marked by "+".

Now, assume that we have the knowledge about the proportions of the three species among all the Iris flowers $(r_1, r_2, r_3)$ and the Iris flowers that need to be classified reflect this composition. For the same $\alpha = 5\%$, $\gamma = 95\%$, $S = 10,000$ and $Q = 10,000$, and with, for example, $(r_1, r_2, r_3) = (0.3, 0.4, 0.3)$, the exact critical constant $\lambda$ that solves Equation (6) is computed by our R program to be $\lambda_{exa} = 7.737$. As expected, $\lambda_{exa}$ is smaller than $\lambda_{con}$ and, as a result, the corresponding confidence set $\mathcal{C}_\mathcal{T}(\mathbf{y})$ in Equation (1) with $\lambda = \lambda_{exa}$ and acceptance sets $\mathcal{A}_l^{exa} = \left\{ \mathbf{y} \in R^p : (\mathbf{y} - \hat{\mu}_l)^T \hat{\Sigma}_l^{-1}(\mathbf{y} - \hat{\mu}_l) \leq \lambda_{exa} \right\}$, $l = 1, 2, 3$, are also smaller than the $\mathcal{A}_l^{con}$ given by Liu et al. (2019).

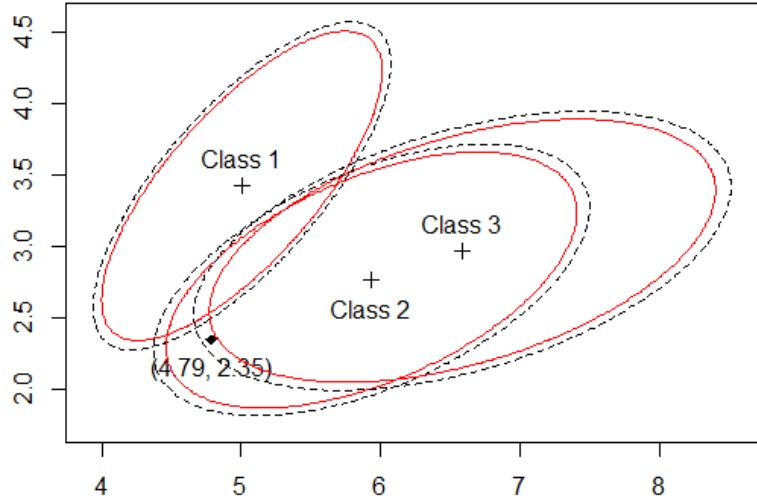

**Figure 1.** The exact (solid) and conservative (dotted) acceptance sets for the three classes.

The acceptance sets $\mathcal{A}_l^{exa}$, $l = 1, 2, 3$ are plotted in Figure 1 by the solid ellipsoidal regions. For example, if a future object has $\mathbf{y} = (4.79, 2.35)$, marked by a solid dot in Figure 1, then the conservative confidence set of Liu et al. (2019) classifies the object as from Classes 2 and 3 since this $\mathbf{y}$ belongs to both $\mathcal{A}_2^{con}$ and $\mathcal{A}_3^{con}$. However, the new exact confidence set of this paper classifies the object as from Class 2 only since this $\mathbf{y}$ belongs to $\mathcal{A}_2^{exa}$ but not $\mathcal{A}_1^{exa}$ or $\mathcal{A}_3^{exa}$. This demonstrates the advantage of the new confidence set using $\lambda_{exa}$ in this paper over the conservative confidence set using $\lambda_{con}$ by Liu et al. (2019). We have also computed the value of $\lambda_{exa}$ for several other given $(r_1, r_2, r_3)$. For example, $\lambda_{exa} = 7.706$ for $(r_1, r_2, r_3) = (1/3, 1/3, 1/3)$, $\lambda_{exa} = 7.865$ for $(r_1, r_2, r_3) = (0.1, 0.45, 0.45)$, and $\lambda_{exa} = 8.019$ for $(r_1, r_2, r_3) = (0.1, 0.7, 0.2)$. The conservative $\lambda_{con} = 9.175$ is considerably, ranging from 14% to 19%, larger than these $\lambda_{exa}$ values.

One can download from http://www.personal.soton.ac.uk/wl/Classification/ the R computer program `ExactConfidenceSetClassifier.R` that implements this simulation method of computing the critical constant $\lambda_{exa}$. The computation of one $\lambda_{exa}$ using $(S, Q) = (10,000, 10,000)$ takes about 13 h on an ordinary Window's PC (Core(TM2) Duo CPU P8400@2.26 GHz).

However, it must be emphasized the new confidence set is valid only if the assumption in Equation (7) is true. If the assumption does not hold, then the conservative confidence set of Liu et al. (2019) should be used in order for the statement in Equation (5) to hold.

## 4. Conclusions

The probability statement in Equation (5) allows that the confidence sets by Liu et al. (2019) have the nice interpretation that, with confidence level $\gamma$ about the randomness in the training dataset $\mathcal{T}$, at least $1 - \alpha$ proportion of the confidence sets $\mathcal{C}_\mathcal{T}(\mathbf{y}_j)$, $j = 1, 2, \ldots$ contain the true classes $c_j$, $j = 1, 2, \ldots$ of the future objects $\mathbf{y}_j$, $j = 1, 2, \ldots$. However, the confidence set given by Liu et al. (2019) is conservative in that the $\lambda$ in the confidence set in Equation (1) is computed to solve the equation in Equation (4), which implies the constraint in Equation (5). This paper considers how to compute the $\lambda$

in the confidence set in Equation (1) so that the probability in Equation (5) is equal to $\gamma$, i.e. from the Equation (6). The confidence sets using the $\lambda$ that solves the Equation (6) have the confidence level equal to $\gamma$ and so are exact. We show that this can be accomplished under the extra assumption given in Equation (7), which may be sensible in some applications.

As the $\lambda_{exa}$ that solves Equation (6) is smaller than the $\lambda_{con}$ that solves Equation (4) used by Liu et al. (2019), the new confidence sets are smaller and so better than the confidence sets given by Liu et al. (2019).

One wonders whether there are other sensible assumptions that allow the $\lambda$ to be solved from Equation (6). This warrants further research.

If $\mathcal{C}_{\mathcal{T}}(\mathbf{y})$ for a future object $\mathbf{y}$ is empty then, since $\mathbf{y}$ must be from one of the $k$ classes, $\mathcal{C}_{\mathcal{T}}(\mathbf{y})$ can be augmented to include the class that has the largest posterior probability using the naive Bayesian classifier as in the work by Liu et al. (2019). The probability statement in Equation (5) clearly holds under this augmentation to $\mathcal{C}_{\mathcal{T}}(\mathbf{y})$ only when $\mathcal{C}_{\mathcal{T}}(\mathbf{y})$ is empty.

There are applications in which information about the proportions $r_l$ would be known with uncertainty. For example, the training set may be a representative sample from the population and as such the proportion of each class can be estimated, or the proportions might have been estimated by a previous independent auxiliary dataset. If one replaces the $r_l$s in Equation (8) by these estimates then the $\lambda$ solved in Equation (8) will depend on these estimates and so be random. As a result, the probability statement in Equation (5) is no longer valid. How to deal with these applications warrants further research.

Finally, the classifier of Liu et al. (2019) is developed from the idea of Lieberman et al. [9,10]. The same idea was also used by, for example, Mee et al. (1991) [11], Han et al. (2016) [12], Liu et al. (2016) [13] and Peng et al. (2019) [14], who all used conservative critical constants as did Liu et al. (2019). The idea of this paper can be applied to all these works to compute exact critical constants under suitable extra assumptions.

**Author Contributions:** methodology, software, F.B., A.J.H., W.L.; software, W.L.

**Acknowledgments:** We would like to thank the referees for critical and constructive comments on the earlier version of the paper.

**Conflicts of Interest:** The authors declare no conflict of interest.

## Appendix A. Mathematical Details

In this appendix, we show the equivalence of Equations (6) and (8) under the assumption in Equation (7). Note first the well known fact (cf. [15]) that $\hat{\boldsymbol{\mu}}_l \sim N(\boldsymbol{\mu}_l, \Sigma_l/n_l)$, $(n_l - 1)\hat{\Sigma}_l = \sum_{m=1}^{n_l-1} \mathbf{z}_{lm}\mathbf{z}_{lm}^T$ with $\mathbf{z}_{l1}, \ldots, \mathbf{z}_{l(n_l-1)}$ being i.i.d. $N(\mathbf{0}, \Sigma_l)$ random vectors independent of $\hat{\boldsymbol{\mu}}_l$.

Among the $N$ future objects that need to be classified, let $N_l$ be the number of objects actually from the $l$th class with the feature measurements denoted as $\mathbf{y}_{l1}, \ldots, \mathbf{y}_{lN_l}$, $l = 1, \ldots, k$. Clearly, we have $N_1 + \cdots + N_k = N$ and

$$
\liminf_{N\to\infty} \frac{1}{N} \sum_{j=1}^{N} I_{\{c_j \in \mathcal{C}_{\mathcal{T}}(\mathbf{y}_j)\}}
$$

$$
= \liminf_{N\to\infty} \frac{1}{N} \sum_{l=1}^{k} \sum_{i=1}^{N_l} I_{\{c_l \in \mathcal{C}_{\mathcal{T}}(\mathbf{y}_{li})\}}
$$

$$
= \liminf_{N\to\infty} \sum_{l=1}^{k} \frac{N_l}{N} \left( \frac{1}{N_l} \sum_{i=1}^{N_l} I_{\{c_l \in \mathcal{C}_{\mathcal{T}}(\mathbf{y}_{li})\}} \right). \tag{A1}
$$

We have from the classical strong law of large numbers (cf. [16]) that

$$\lim_{N_l \to \infty} \frac{1}{N_l} \sum_{i=1}^{N_l} \left[ I_{\{c_l \in \mathcal{C}_\mathcal{T}(\mathbf{y}_{li})\}} - E_{\mathbf{y}_{li}|\mathcal{T}} I_{\{c_l \in \mathcal{C}_\mathcal{T}(\mathbf{y}_{li})\}} \right] = 0, \tag{A2}$$

in which the conditional expectation $E_{\mathbf{y}_{li}|\mathcal{T}}$ is used since all the confidence sets $\mathcal{C}_\mathcal{T}(\mathbf{y}_{li})$ $(i = 1, \ldots, N_l)$ use the same training dataset $\mathcal{T}$. By noting that $\mathbf{y}_{li}, i = 1, \ldots, N_l$ are from the $l$th class and thus have the same distribution $N(\boldsymbol{\mu}_l, \Sigma_l)$, we have from the definition of $\mathcal{C}_\mathcal{T}(\mathbf{y})$ in Equation (1) that

$$
\begin{aligned}
&E_{\mathbf{y}_{li}|\mathcal{T}} I_{\{c_l \in \mathcal{C}_\mathcal{T}(\mathbf{y}_{li})\}} \\
={}& P_{\mathbf{y}_{l1}|\mathcal{T}} \{c_l \in \mathcal{C}_\mathcal{T}(\mathbf{y}_{l1})\} \\
={}& P_{\mathbf{y}_{l1}|\mathcal{T}} \left\{ (\mathbf{y}_{l1} - \hat{\boldsymbol{\mu}}_l)^T \hat{\Sigma}_l^{-1} (\mathbf{y}_{l1} - \hat{\boldsymbol{\mu}}_l) \le \lambda \right\} \\
={}& P_{\mathbf{w}_l \mid \mathbf{u}_l, \{\mathbf{v}_{lm}\}} \left\{ (\mathbf{w}_l - \mathbf{u}_l)^T \left( \frac{1}{n_l - 1} \sum_{m=1}^{n_l - 1} \mathbf{v}_{lm} \mathbf{v}_{lm}^T \right)^{-1} (\mathbf{w}_l - \mathbf{u}_l) \le \lambda \right\}
\end{aligned}
\tag{A3}
$$

where

$$
\begin{aligned}
\mathbf{w}_l &= \Sigma_l^{-1/2} (\mathbf{y}_{l1} - \boldsymbol{\mu}_l) \sim N(\mathbf{0}, I_p) \\
\mathbf{u}_l &= \Sigma_l^{-1/2} (\hat{\boldsymbol{\mu}}_l - \boldsymbol{\mu}_l) \sim N(\mathbf{0}, I_p / n_l) \\
\mathbf{v}_{lm} &= \Sigma_l^{-1/2} \mathbf{z}_{lm} \sim N(\mathbf{0}, I_p), \ m = 1, \cdots, n_l - 1
\end{aligned}
$$

with all the $\mathbf{w}_l$s, $\mathbf{u}_l$s and $\mathbf{v}_{lm}$s being independent. Note that $\mathbf{w}_l$ depends on the future observation $\mathbf{y}_{l1}$ but not the training dataset $\mathcal{T}$, while $\mathbf{u}_l$ and $\{\mathbf{v}_{lm}\}$ depend on the training dataset $\mathcal{T}$ but not the future observations.

Combining the assumption in Equation (7) with Equations (A1)–(A3) gives

$$\liminf_{N \to \infty} \frac{1}{N} \sum_{j=1}^{N} I_{\{c_j \in \mathcal{C}_\mathcal{T}(\mathbf{y}_j)\}} = \sum_{l=1}^{k} r_l P_{\mathbf{w}_l \mid \mathbf{u}_l, \{\mathbf{v}_{lm}\}} \left\{ (\mathbf{w}_l - \mathbf{u}_l)^T \left( \frac{1}{n_l - 1} \sum_{m=1}^{n_l - 1} \mathbf{v}_{lm} \mathbf{v}_{lm}^T \right)^{-1} (\mathbf{w}_l - \mathbf{u}_l) \le \lambda \right\},$$

from which the equivalence of Equations (6) and (8) follows immediately.

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
