# Peer review of "Confidence Sets for Statistical Classification (II): Exact Confidence Sets"

_stats, doi:10.3390/stats2040030_

Round 1

Reviewer 1 Report

The paper considers the construction of confidence sets for the classification of observations from two or more multivariate normal distributions based on a training set of data points. This paper extends the work in Liu et al 2019 by developing an exact confidence set with tighter bounds in the case where the proportion of future observations originating from each class may be assumed to be known a priori. The paper is well written and the results seem to be correctly derived and implemented.   Main comments:   1. I think the assumption that there are situations where it is reasonable to assume the proportion of subjects from each class is known a priori. However, in other cases such information would be known with uncertainty. In particular, the training set may be a representative sample from the population and as such the proportion of each class can be estimated. Similarly the proportions might have been estimated by a previous independent auxiliary dataset. It would be helpful for the conclusion to briefly discuss how difficult it would be to accommodate these situations into the computation of \lambda.   2. While having a smaller \lambda results in confidence sets with exact coverage, it does have the by-product of giving a higher proportion of cases where the classifier set is the empty set. It would be helpful to report, for the example in Section 3, the proportion of times an observation would be classified to the empty set for future data assuming the MVN distributions from the training data.   Minor corrections:   P1 Abstract L9: As the the exact -> As the exact
P12 L1: Due CPU -> Duo CPU

Author Response

We thank the referees for constructive comments. Attachment is our responses and changes in response to the referees' comments.

Reviewer 2 Report

as I can understand, authors use Monte-Carlo to estimate quantile for Mahalanobis distance with sample mean and covariance used (Eq.1). But the distance has exact distribution, see for example Hardin, Johanna, and David M. Rocke. 2005. “The Distribution of Robust Distances.” Journal of Computational and Graphical Statistics 14 (4): 928–46. doi:10.1198/106186005X77685, section 3. It could be worth to compare results from simulation with theoretical one.

Author Response

(The authors gave the same response as above.)

Reviewer 3 Report

The paper is very interesting.

The paper is continuation of the authors previous papers. The authors describe their new algorithm, that is used for one sample data. If somebody need to analyze another type of data, is it possible to adapt the algorithm and to use it? Computationally complexity will be better or worse? Who can use this algorithm?

Author Response

(The authors gave the same response as above.)
